# Shaping Belief States with Generative Environment Models for RL

**Karol Gregor**  **Danilo Jimenez Rezende**  **Frederic Besse**

**Yan Wu**  **Hamza Merzic**  **Aäron van den Oord**

**Google DeepMind**
London, UK
{karolg, danilor, fbesse, yanwu, hamzamerzic, avdnoord}@google.com

## Abstract

When agents interact with a complex environment, they must form and maintain beliefs about the relevant aspects of that environment. We propose a way to efficiently train expressive generative models in complex environments. We show that a predictive algorithm with an expressive generative model can form stable belief-states in visually rich and dynamic 3D environments. More precisely, we show that the learned representation captures the layout of the environment as well as the position and orientation of the agent. Our experiments show that the model substantially improves data-efficiency on a number of reinforcement learning (RL) tasks compared with strong model-free baseline agents. We find that predicting multiple steps into the future (overshooting), in combination with an expressive generative model, is critical for stable representations to emerge. In practice, using expressive generative models in RL is computationally expensive and we propose a scheme to reduce this computational burden, allowing us to build agents that are competitive with model-free baselines.

## 1  Introduction

We are interested in making agents that can solve a wide range of tasks in complex and dynamic environments. While tasks may be vastly different from each other, there is a large amount of structure in the world that can be captured and used by the agents in a task-independent manner. This observation is consistent with the view that such general agents must understand the world around them [1]. The collection of algorithms that learn representations by exploiting structure in the data that are general enough to support a wide range of downstream tasks is what we refer to as unsupervised learning or self-supervised learning. We hypothesize that an ideal unsupervised learning algorithm should use past observations to create a stable representation of the environment. That is, a representation that captures the global factors of variation of the environment in a temporally coherent way. As an example, consider an agent navigating in a complex landscape. At any given time, only a small part of the environment is observable from the the perspective of the agent. The frames that this agent observes can vary significantly over time, even though the global structure of the environment is relatively static with only a few moving objects. An useful representation of such an environment would contain, for example, a map describing the overall layout of the terrain. Our goal is to learn such representations in a general manner.

Predictive models have long been hypothesized as a general mechanism to produce useful representations based on which an agent can perform a wide variety of tasks in partially observed worlds

[2, 3]. A formal way of describing agents in partially observed environments is through the notion of partially observable Markov decision processes [4, 5] (POMDPs). A key concept in POMDPs is the notion of a *belief-state*, which can be defined as the sufficient statistics of future states [6, 7, 8]. In this paper we refer to belief-states as any vector representation that is sufficient to predict future observations as in [9, 10].

A fundamental problem in building useful environment models, which we want to address in this work, is long-term consistency [11, 12]. This problem is characterized by many models' failure to perform coherent long-term predictions, while performing accurate short-term predictions, even in trivial but partially observed environments [11, 13, 14].

We argue that this is not merely a model capacity issue. As previous work has shown, globally coherent environment maps can be learned by position conditioned models [15]. We thus propose that this problem is better diagnosed as the failure in model conditioning and a weak objective, which we discuss in more details in Section 2.1.

The main contributions of this paper are: 1) We demonstrate that an expressive generative model of rich 3D environments can be learned from merely first-person views and actions to capture long-term consistency; 2) we provide an analysis of three different belief-state architectures (LSTM [16], LSTM + Kanerva Memory [17] and LSTM + slot-based Memory [18]) on the ability to decode the environment's map and the agent's location. 3) we design and perform experiments to test the effects of both overshooting and memory, demonstrating that generative models benefit from these components more than deterministic models; 4) we show that training agents that share their belief-state with the generative model have substantially increased data-efficiency compared to a strong model-free baseline [19, 18], without significantly affecting the training speed; 5) we show one of the first agents that learns to collect construction materials in order to build complex structures from a first-person perspective in a 3d environment.

The remainder of this paper is organized as follows: in Section 2 we introduce the main components of our agent's architecture and discuss some key challenges in using expressive generative models in RL. Namely, the problem of conditioning, Section 2.1, and why next-step models are not sufficient in Section 2.2, in Section 3 we discuss related research. Finally, we describe our experiments in Section 4.

## 2 Methods

In this section we describe our proposed agent and model architectures. Our agent has two main components. The first is a recurrent neural network (RNN) as in [19] which observes frames $x_t$, processes them through a feed-forward network and aggregates the resulting outputs by updating its recurrent state $b_t$. From this updated state, the agent core then outputs the action logits, a sampled action and the value function baseline. The second component is the unsupervised model, which can be: (i) a contrastive loss based on action-conditional CPC [20]; (ii) a deterministic predictive model (similar to [11]) and (iii) an expressive generative predictive model based on ConvDRAW, [21]. We also investigate different memory architectures in the form of slot-based memory (as used in the reconstructive memory agent, RMA) [18] and compressive memory (Kanerva) [17]. The unsupervised model consists of a recurrent network, which we refer to as simulation network or *SimCore*, that starts with a given belief state $s_t^0 = b_t$ at some random time $t$, and then simulates deterministically forward, seeing only the actions of the agent. After simulating for $k$ steps, we use the resulting state $s_t^k$ to predict the distribution of frames $p(x_{t+k}|b_t, a_{t...(t+k)}) = p(x_{t+k}|s_t^k)$(in cases (ii) and (iii)). A diagram illustrating this agent is provided in Figure 1 and the precise computation steps are provided in Table 1. A concrete example of the computation of the model's loss is provided as pseudo-code in Appendix K.

Evaluating the loss of an expressive generative model for an entire sequence is computationally expensive. We address this by only computing the model's loss at a small random subset of the frames in the sequence as shown in Table 1.

### 2.1 Frame generative models and the problem of conditioning.

It is known that expressive frame generative models are hard to condition [22, 23, 24, 25]. This is especially problematic for learning belief-states, because it is this conditioning that provides the learning

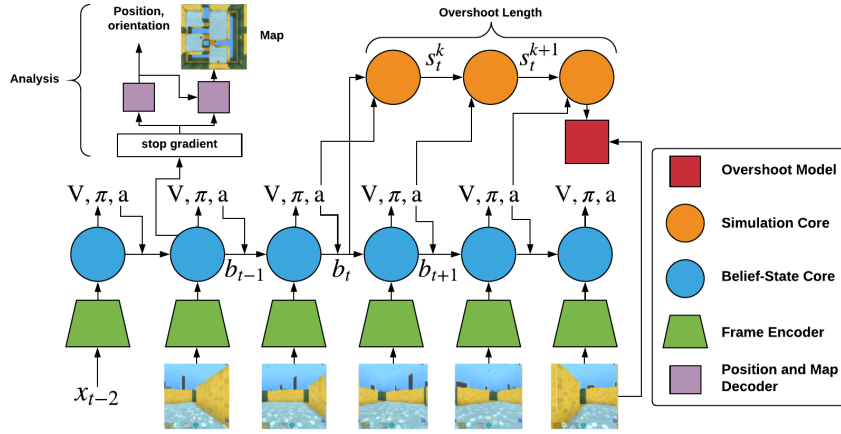

Figure 1: Diagram of the agent and model. The agent receives observations $x$ from the environment, processes them through a feed-forward residual network (green) and forms a state using a recurrent network (blue), online. This state is a belief state and is used to calculate policy and value as well as being the starting point for predictions of the future. These are done using a second recurrent network (orange) - a simulation network (SimCore) that simulates into the future seeing only the actions. The simulated state is used to conditioning for a generative model (red) of a future frame.

| | | | |
|---|---|---|---|
| **Agent Core** | Belief State Update | $b_t$ | $= \text{RNN}_{\text{agent}}(b_{t-1}, a_{t-1}, x_t)$ |
| | Value and Policy Logits | $V_t, o_t$ | $= f(b_t)$ |
| | Action | $a_t$ | $\sim \text{Cat}(o_t)$ |
| **SimCore** | Simulation State Initialization | $s_t^0$ | $= b_t$ |
| | Simulation State Update | $s_t^{k+1}$ | $= \text{RNN}_{\text{simcore}}(s_t^k, a_{t+k})$ |
| | Simulation Starting Times | $t_i$ | $\sim \text{unif}(1, L_u)$ |
| | Likelihood Evaluation Times | $\delta_k^i$ | $\sim \text{unif}(1, \min(L_u, t_i + L_o))$ |
| | Predictive Loss | $\mathcal{L}$ | $= -\sum_{i=1}^{N_g} \sum_{k=1}^{N_t} \log p(x_{t_i + \delta_k^i} \mid s_{t_i}^{\delta_k^i})$ |

Table 1: Agent and simulation core definitions. The agent's RNN state $b_t$ is what we call the belief state. At the start of a simulation, the SimCore's RNN state $s_t^k$ is initialized to be equal to the belief state $s_t^0 = b_t$. The states of the SimCore RNN $s_t^k$ are used at times $\delta_k^i$ to condition the model of the frames at times $t_i + \delta_k^i$. Here $L_u$ is the total length of an episode, $N_g$ (typically 2) is the number of points in the future used to evaluate the predictive loss, $N_t$ (typically 6) is the number of random points along the trajectory where we unroll the predictive model and $L_o$ is the overshoot length (typically 4-32), which is the maximum time-length used to train the predictive model. We choose $N_g$ and $N_t$ small compared to $L_o$ to maintain a low computational cost of the model's loss.

signal for the formation of belief-states. If the generative model ignores the conditioning information, it will not be possible to optimize the belief-states. More precisely, if the generative model fails to use the conditioning we have $\mathcal{L} = -\sum_{i=1}^{N_t} \sum_{k=1}^{N_g} \ln p(x_{t+\delta_k} \mid s_{t_i}^{\delta_k}) \approx -\sum_{i=1}^{N_t} \sum_{k=1}^{N_g} \ln p(x_{t+\delta_k})$ and thus $\nabla_{b_t} \mathcal{L} \approx 0$, and consequently learning the belief-state $b_t$ is not possible.

We observe this problem by experimenting with expressive state of-the-art deep generative models such as PixelCNN [26], ConvDRAW [21] and RNVP [27]. We found empirically that a modified ConvDRAW in combination with GECO [28] works well in practice, which allows us to learn stable belief-states while maintaining good sample quality. As a result, we can use our model to consistently simulate many steps into the future. More details of the modified ConvDRAW architecture and GECO optimization experiments are provided in Appendix A and Appendix G respectively.

## 2.2 Why is next-step prediction not sufficient?

A theoretically sufficient and conceptually simple way to form a belief state $b_t$ is to train a next-step prediction model $p(x_{t+1} \mid x_1, \ldots, x_t) = p(x_{t+1} \mid b_t)$, where $b_t = \text{RNN}(b_{t-1}, x_t, a_t)$ summarizes the past. Under an optimal solution, it contains all the information needed to predict the future

$p(x_{t+1}, x_{t+2}, \ldots | b_t)$ because any joint distribution can be factorized as a product of such conditionals: $p(x_{t+1}, x_{t+2}, \ldots | b_t) = p(x_{t+1} | b_t) \times p(x_{t+2} | b_{t+1} = f(x_{t+1}, b_t)) \times \ldots$ This reasoning motivates a lot of research using next-step prediction in RL, e.g. [29, 30].

We argue that next-step prediction exacerbates the conditioning problem described in Section 2.1. In a physically realistic environment the immediate future observation $x_{t+1}$ can be predicted, with high accuracy, as a near-deterministic function of the immediate past observations $x_{(t-\kappa),\ldots,t}$. This intuition can be expressed as $p(x_{t+1} | x_{(t-\kappa),\ldots,t}, a_{(t-\kappa-1),\ldots,(t-1)}, b_{(t-\kappa)}) \approx p(x_{t+1} | x_{(t-\kappa),\ldots,t}, a_{(t-\kappa-1),\ldots,(t-1)})$. That is, the immediate past weakens the dependency on belief-state vectors, resulting in $\nabla_{b_t} \mathcal{L} \approx 0$. Predicting the distant future, in contrast, requires knowledge of the global structure of the environment, encouraging the formation of belief-states that contain that information.

Generative environment models trained with overshooting have been explored in the context of model-based planning [31, 32, 33, 34]. But evidence of the effect of overshooting has been restricted to the agent's performance evaluation [33, 31]. While there is some evidence that overshooting improves the ability to predict the long-term future [11], there is no extensive study examining which aspects of the environment are retained by these models.

As noted above, for a given belief-state the entropy of the distribution of target observations increases with the overshoot length (due to partial observability and/or randomness in the environment), going from a near deterministic (uni-modal) distribution to a highly multi-modal distribution. This leads us to hypothesize that deterministic prediction models should benefit less from growing the overshooting length compared to generative prediction models. Our experiments below support this hypothesis.

### 2.3 Belief-states, Localization and Mapping

Extracting a consistent high-level representation of the environment such as the bird's eye view "map" from merely first-person views and actions in a completely unsupervised manner is a notoriously difficult task [12] and a lot of the success in addressing this problem is due to the injection of a substantial amount of human prior knowledge in the models [35, 36, 37, 38, 39, 40].

While previous work has primarily focused on extracting human-interpretable maps of the environment, our approach is to decode position, orientation and top down view or layout of the environment from the agent's belief-state $b_t$. This decoding process does not interfere with the agent's training, and is not restricted to 2D map-layouts.

We use one-layer MLP to predict the discretized position and orientation and a convolutional network to predict the top down view (map decoder). When the belief-state $b_t$ is learned by an LSTM, it is composed of the LSTM hidden state $h_t$ and the LSTM cell state $c_t$. Since the location and map decoder need access to the full belief-state, we condition these maps on the vector $u_t = \text{concat}(h_t, \tanh(c_t))$. When using the episodic, slot based, RMA memory we first take a number of reads from the memory conditioned on the current belief-state $b_t$ and concatenate them with $u_t$ defined above. For the Kanerva memory we learn a fixed set of read vectors and concatenate the retrieved memories with $u_t$.

## 3 Other Related Work

The idea of learning general world models to support decision-making is probably one of the most pervasive ideas in AI research, [30, 11, 41, 42, 14, 43, 44, 45, 46, 33, 30, 47, 48, 3, 49, 50, 51, 52]. In spite of a vast literature supporting the potential advantages of model-based RL, it has been a challenge to demonstrate the benefits of model-based agents in visually rich, dynamic, 3D environments. The challenge of model-based RL in rich 3D environments has compelled some researchers to use privileged information such as camera-locations [15], depth information [53], and other ground-truth state variables of the state simulator [54, 49]. On the other hand, some work has provided evidence that we may not need very expressive models to benefit to some degree [30].

Our proposed model could in principle be used in combination with planning algorithms. But we take a step back from planning and focus more on the effect of various model choices on the learned representations and belief states. This approach is similar to having a representation shaped via auxiliary unsupervised losses for a model-free RL agent. Combining auxiliary losses with

reinforcement learning is an active area of research and a variety of auxiliary losses have been explored. A non-exhaustive list includes pixel control [55], contrastive predictive coding (CPC) [56], action-conditional CPC [20], frame reconstruction [35, 57], next-frame prediction [18, 58, 20, 59] and successor representations [60, 61, 62].

As in [20, 10, 50] our proposed architecture has a shared belief-state between the generative model and the agent's policy network. The closest paper to our work is [20], where a comparison is made between action-conditional CPC and next-step prediction using a deterministic next-step model. There are several key differences between this paper and our work: (i) We analyze the decoding of the environment's map from the belief state; (ii) We show that while next-frame prediction may be sufficient to encode position and orientation, it is necessary to combine expressive generative models with overshoot to form a consistent map representation; (iii) We demonstrate that an expressive generative model can be trained to simulate visually rich 3D environments for several steps in the future; (iv) We also analyze the impact on RL performance of various model choices. We also discuss and propose solutions to the general problem of conditioning expressive generative models.

# 4 Experiments

We analyze our agent's performance with respect to both its ability to represent the environment (e.g. knows its position and map layout) and RL performance. Our experiments span three main axes of variation: (i) the choice of unsupervised loss for the overshoots (e.g. deterministic prediction, generative prediction and contrastive); (ii) the choice of overshoot length and (iii) the choice of architecture for the belief-state and simulation RNNs (LSTM [16], LSTM with Kanerva memory [17] and LSTM with the memory from the reconstructive memory agent (RMA) [18]). RMA uses a slot based memory that stores all past vectors, whereas Kanerva memory updates existing memories with new information in a compressive way, see Appendix B for more details.

The agent is trained using the IMPALA framework [19], a variant of policy gradients, see Appendix D for details. The model is trained jointly with the agent, sharing the belief network. We find that the running speed decreases only by about $20 - 40\%$ compared to the agent without model. We use Adam for optimization [63]. The detailed choice of various hyperparameters is provided in Appendix F.

Our experiments are performed using four families of procedural environments: (a) DeepMind-Lab levels [64] and three new environments that we created using the Unity Engine: (b) Random City; (c) Block building environment; (d) Random Terrain.

## 4.1 Random City

The Random City is a procedurally generated 3D navigation environment, Figure 2 showing first person view (top row) and the top down view (second row). At the beginning of an episode, a random number of randomly colored boxes (i.e. "buildings") are placed on top of a 2d plane. We used this environment primarily as a way to analyze how different model architectures affect the formation of consistent belief-states. We generated training data for the models using fixed handcrafted policy that chooses random locations and path planning policy to move between these locations and analyze the model and the content of the belief state (no RL in this experiment).

In the third row of Figure 2 we show the top down view reconstructions from the belief state (to emphasize, the belief-state was not trained with this information). We see that whenever the agent sees a new building, the building appears on a map, and it still preserves the other buildings seen so far even if they are not in its current field-of-view. Rows four and five show a later part of an input sequence (when the model has seen a large part of the environment) and a rollout from the model. We see that the model is able to preserve the information in its state and use this information to correctly simulate forward.

We analyze the effects of self-supervised loss type, overshoot length and memory architecture on position prediction and map decoding accuracy. The results are shown in Figure 3. We make the following observations: (i) an increase in the overshoot length improves the ability to decode the agent's position and the map layout for all losses (up to certain length); (ii) The contrastive loss provides the best decoding of the agent's position for all overshoot lengths Figure 3a; (iii) The generative prediction loss provides the best map decoding and is the most sensitive to the overshoot

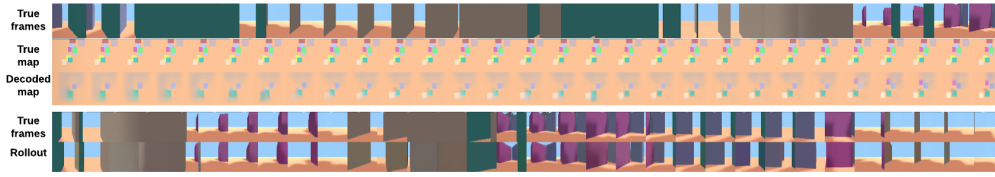

Figure 2: Random City environment. **Rows**: 1. Input to the model sequence starting from the beginning of the episode. 2. Top down view (a map). 3. Top down view decoded from the belief state. The belief state was not trained with this decoding signal, but only from the first person view (top row). We see that the model is able to fill up the map as it sees new frames. 4. Frames later in the sequence (after 170 steps). 5. Rollout from the model. The model knows what it will see as the agent rotates. See supplementary video `https://youtu.be/d0nvAp_wxv0`.

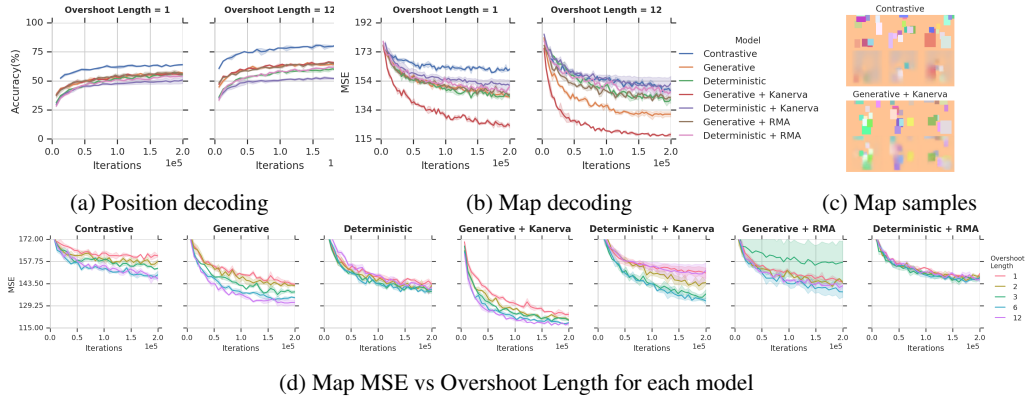

(a) Position decoding          (b) Map decoding          (c) Map samples

(d) Map MSE vs Overshoot Length for each model

Figure 3: The choice of model and overshoot length have significant impact on state representation. **(a)** All models benefit from an increase in the overshoot length with respect to position decoding, with the Contrastive model reaching higher accuracy; **(b)** The Generative models are the most sensitive to overshoot length with respect to Map decoding MSE. A substantial reduction in map decoding MSE is obtained by using architectures with memory; **(c)** Examples of decoded maps. Each block shows real maps (top-row) and decoded maps (bottom-row). Top block: Contrastive model samples at Overshoot Length 1 (MSE of approx. 160); Bottom block: Generative + Kanerva at Overshoot Length 12 (MSE of approx. 117). We can clearly notice the difference in the details for both models. **(d)** Effect of overshoot on environment's map decoding. This analysis shows that Generative and Generative + Kanerva benefit the most from an increase in overshoot length in contrast to Deterministic and Contrastive architectures. In particular, we observe that Generative + Kanerva architecture is particularly good at forming belief-states that contain a map of the environment.

length with respect to map decoding error Figure 3d. (iv) The combination of generative model with Kanerva memory provides the best map decoding accuracy.

We also see that the contrastive loss is very good at localization but poor at mapping. This loss is trained to distinguish a given state from others within the simulated sequence and from other elements of the mini-batch. We hypothesize that keeping track of location very accurately allows to distinguish a given time point from others, but that in a varied environment it is easy for the network to distinguish one batch element from others without forming a map of the environment.

We also see that Kanerva memory works significantly better then pure LSTM and the slot based memory. However, the latter result might be due to limitation of the method used to analyze the content of the belief state. In fact it is likely that the information is in the state since slot based memory stores all the past vectors, but that it is hard to extract this information. This also raises an interesting point - what is a belief state? Storing all past data contains all the information a model can have. We suggest that what we are after is a more compact representation that is stored in a easy to access way. Kanerva memory aims to not only to store the past information but integrate it with already stored information in a compressed way.

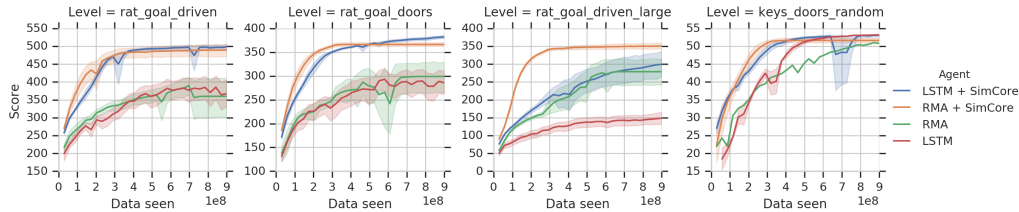

Figure 4: Generative SimCore results in substantial data-efficiency gains for agents in DeepMind-Lab relative to a strong model-free baseline. We also observe that model-free agents have substantially higher variance in their scores. See supplementary video `https://youtu.be/d0nvAp_wxv0`.

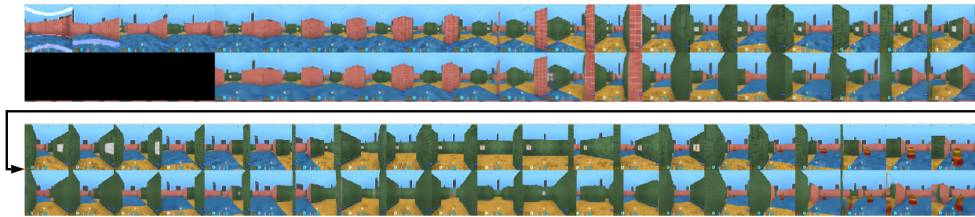

Figure 5: The input and the rollout in DeepMind Lab. The agent is able to correctly rollout for many steps, and remember where the rewarding object is (the object in the bottom right frames).

## 4.2 DeepMind Lab

DeepMind Lab [64] is a collection of 3D, first-person view environments where agents perform a diverse set of tasks. We use a subset of DeepMind Lab (rat_goal_driven, rat_goal_doors, rat_goal_driven_large and keys_doors_random) to investigate how the addition of the generative prediction loss with overshoot affects the agent's representation or belief-state as well as its RL performance.

We compare four agents in the following experiments. The first termed LSTM is the standard IMPALA agent with LSTM recurrent core. Next agent, termed RMA is the agent of [18], the core of which consist of and LSTM and a slot based episodic memory. The final two agents termed LSTM+SimCore and RMA+SimCore are the same as LSTM and RMA agents, but with the model loss added.

The results of our experiments are shown in Figure 4. We see that adding the model loss improves performance for both LSTM and RMA agents. While Kanerva memory significantly helps in the data regime we found it to be unstable in the RL setting. More work is required to solve this problem. We found that using the RMA memory helped substantially with large environments as shown in Figure 4(rat_goal_driven_large).

We found that map reconstruction loss varies significantly during training. This could be due to policy gradients affecting the belief state, changing policy or changing of the way the model represents the information, with decoder having hard time keeping up. We found that longer overshoot lengths perform better than shorter ones, but that did not translate into improved RL performance. This could also be an artifact of the environment - there are permanent features present on the horizon, and the agent does not need to know the map to navigate to the goal. The model is able to correctly rollout for a number of steps, Figure 5 knowing where the rewarding object is (the object on the bottom right).

## 4.3 Voxel environment

We want to create an environment that requires agents to learn complex behaviours to solve tasks. For this, we created a voxel-based, procedural environment with Unity that can be modified by the agents via building mechanisms, resulting in a large combinatorial space of possible voxel configurations and behavioural strategies. See accompanying video for examples of this environment and of learned behaviors.

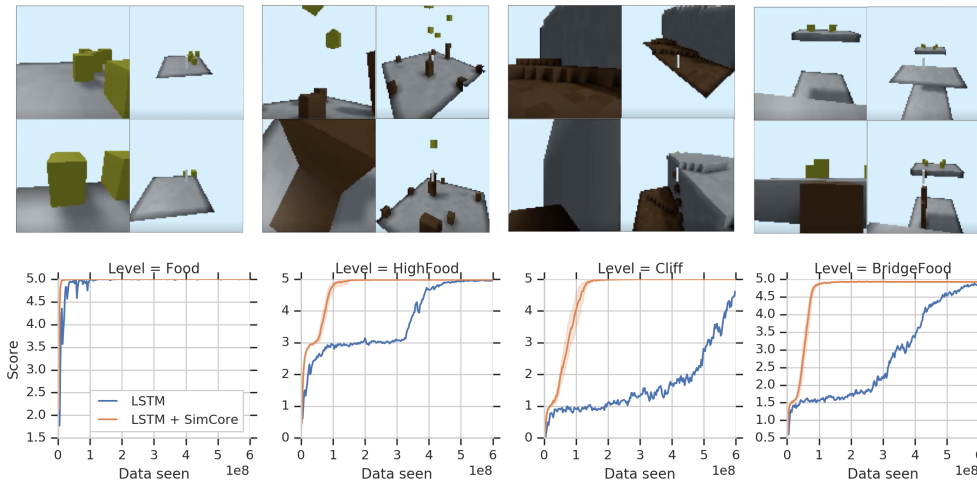

Figure 6: Top: Voxel levels. There are four levels: BridgeFood, Cliff, Food and HighFood. For each level, four views are shown: Early frame first person view, early frame third person view, later frame first person view, later frame third person view. The agent sees only the first person view and its goal is to pick up yellow blocks, which it needs to get to. The agent has blocks that it can place. The agent learns how to build towers (BridgeFood and HighFood) and stairs (Cliff) to climb to the food. Bottom: Training the agent with SimCore substantially increases data-efficiency. See supplementary video https://youtu.be/dOnvAp_wxv0.

The environment consists of blocks of different types and appearances that are placed in a three dimensional grid. The agent moves continuously through the environment, obeying Unity engine's physics. The agent has a simple inventory, can pick up blocks of certain types, place them back into the world and interact with certain blocks. We build four levels, Figure 6 top, where the goal is to consume all yellow blocks ('food'). The levels are: **Food**: Five food blocks placed at random locations in a plane - this is a curriculum level for the agent to quickly learn that yellow blocks give reward. **HighFood**: The same setting, but the food is also placed at random height. If the food is slightly high, the agent needs to look up and jump to get it. If the food is even higher, the agent needs to place blocks on the ground, jump on them, look up at the food and jump. **Cliff**: There is a cliff of random height with food at the top. The agent needs to first pick up blocks and then build structures to climb and reach the top of the cliff. Interestingly, the agent learns to pick them up and build stairs on the side of the cliff. **Bridge**: The agent needs to get to the other side of a randomly sized gap, either by building a bridge or falling down and then building a tower to climb back up. The agent learns the latter. We also trained the agent on more complex versions of the levels, showing rather competent abilities of building high structures climbing, see Appendix J and the accompanying video.

We compared the LSTM and LSTM+SimCore agents (without an episodic memory) on these levels. In this case, one agent is playing all four levels at the same time. From Figure 6 we see that the SimCore significantly improves the performance on all the levels. In addition we found that the performance is much less sensitive to Adam hyper-parameters as well as unroll length. We also found that the model is able to simulate its movement, building behaviours and block picking, see (Appendix J) for samples.

Finally we tested a map building ability in a more naturalistic, procedurally generated terrain, Figure 11. This environment is harder than the city, because it takes significantly more steps to cross the environment. We also analyzed a simple RL setting of picking up randomly placed blocks. We found that an LSTM agent contains an approximate map, but the information not seen for a while gradually fades away. We hope to scale up these experiments in the future.

# 5  Discussion

In this paper we introduced a scheme to train expressive generative environment models with RL agents. We found that expressive generative models in combination with overshoot can form stable belief states in 3D environments from first person views, with little prior knowledge about the structure of these environments. We also showed that by sharing the belief-states of the model with the agent we substantially increase the data-efficiency in a variety of RL tasks relative to strong baselines. There are more elements that need to be investigated in the future. First, we found that training the belief state together with the agent makes it harder to either form a belief state or decode the map from it. This could result from the effect of policy gradients or changing of policy or changing the way the belief is represented. Additionally we aim towards scaling up the system, either through better training or better use of memory architectures. Finally, it would be good to use the model not only for representation learning but for planning as well.

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
