[Supplementary Material · Shaping_Belief_States_with_Generative_Environment_Models_for_RL_app.pdf]

## Appendix A   Modified ConvDRAW

Convolutional DRAW [21] is an instance of a deep variational auto-encoder [63, 64]. It has a recurrent encoder and recurrent decoder with latent variables at each step, forming auto-regressive distributions over the latent variables, Figure 7.

The decoder has a special layer named canvas, which accumulates the result into distribution over inputs $p(x|z)$. All the operations are convolutional, and all the states are three dimensional (spatial times feature dimensions).

We introduce several modifications to the original formulation. First, we replaced the LSTM recurrence by a product of tanh and sigmoid non-linearity, which halves the number of operations, given the number of feature maps. This is part of the LSTM operation and was also used in [26]. Second, instead of passing a conditioning vector into the decoder, we create a separate network that defines the prior over the latent variables. We find that this helps with conditioning. Finally, we adaptively scale the input loss relative to the latent loss so as to achieve a set target accuracy on the reconstructions, as described in [28].

Figure 7: Diagram of ConvDraw's likelihood computation.

We use 8 iterations of the repeated operations (two are displayed in Figure 7). The input of size $64 \times 64$ is processed through two layer convolutional network with rectified linear units, with hidden sizes $32 \times 32 \times 16$ and $16 \times 16 \times 16$. All the circles have have tanh sigmoid nonlinearity. The encoder recurrent state (blue circles) is of size $8 \times 8$ (after tanh sigmoid multiplication) and the other recurrent parts (orange and red circles) are of size $8 \times 8 \times 64$. The decoder convolutional network hidden layers are of size $16 \times 16 \times 32$ and $32 \times 32 \times 32$. The canvas is the size of the image and contains the current reconstruction. We don't model the variance. The error vector $E$ is the difference between the current canvas and the reconstruction. The kernel sizes of the convolutions between layers that change stride are $4 \times 4$ and between those that don't are $5 \times 5$. The first conditioning operation (the first red circle) is applied four times before producing the first prior on $z$. No weights are shared.

## Appendix B   Memory Architectures

Here we discuss the architectures used to aggregate the observations as well as perform prediction. Such networks need to store a new information quickly and incorporate it into their belief state. There are two places where recurrent networks store information - in activations and in weights. For classic recurrent networks, the weights are updated by back-propagation, which is a slow process that results in small updates at every time step. Therefore such networks need to form the belief state in activations. For this type of network in our experiments, we use the standard LSTM [16].

We use two models that were developed as one shot memory architectures. The first one is that of [18]. It is a slot based memory that at a given time step, takes a specific vector produced from the input and an LSTM, and stores that vector in a new slot in memory. The memory starts empty at the beginning of the episode and a new slot is allocated at every time step. Reading from memory is done using an attention based mechanism. The advantage of such memory is that nothing is forgotten

and one does not need to learn how to write into the memory. The disadvantage is that new slots are being allocated and written to even if nothing new is happening.

More appealing would be a mechanism that compares the current input to what is already stored in the memory, and only makes updates necessary to incorporate the new information present in such input. For such mechanism we use Kanerva machines [17], specifically the version in [65]. The Kanerva machine is a generative model of exchangeable observations, in which the global latent variable is used as a memory. Due to its statistical interpretation, writing into the memory is equivalent to inferring the posterior distribution of this global latent variable given a new observation. This inference is exact and efficient, since the linear Gaussian model underlying a Kanerva machine is analytically tractable.

For all the types of RNNs (LSTM, LSTM+RMA, LSTM+Kanerva), we use the same network architecture for both the belief state and the simulation networks. For the networks with memory, we turn off writing into the memory in the simulation network. The LSTM's in all the cases do not share weights between the belief and simulation networks.

## Appendix C   Integrating Kanerva Machines with the RNNs

This section describes how to integrate a Kanerva Machine (KM) with an RNN, so it functions as an external memory for the SimCore (LSTM + Kanerva in the main text). KMs were originally proposed as unsupervised models trained with lower-bounds of log-likelihoods [17, 65]. Here we present a simplification that works as black-box module, which is trained end-to-end by back-propagation with the rest of the model without incurring any auxiliary loss function.

We use the dynamic version as in [65], which uses fully content-based addressing, requiring only a word $z$ as the input for either reading or writing. The optimal location for memory access is obtained by solving a least-squares problem involving both $z$ and memory mean matrix $M$ [65]. At time step $t$, the memory module takes as input the RNN's previous output $b_{t-1}$, and current action $a_t$ and embedding of the observation $e_t$. They are concatenated and linearly projected to obtain a read word $z_r = L \cdot \text{concat}(b_{t-1}, a_t, e_t)$. $z_r$ is used to query the memory for a read-out, which is then use as an input to the RNN, together with $a_t$ and $x_t$, for updating to $b_t$. To update the memory, we obtain a write word by passing through an MLP a similar concatenated vector using the new state $b_t$: $z_w = \text{MLP}(\text{concat}(b_t, a_t, e_t))$ The memory is updated as the posterior distribution conditioned on $z_w$. These operations are illustrated in Figure 8.

Figure 8: Illustration of a dynamic Kanerva machine integrated with an RNN.

## Appendix D   Reinforcement learning framework

For all reinforcement learning experiments, we assume a standard setup where the agents interact with the environment in discrete time steps and using a discrete set of actions. At a particular time step $t$

the agent with belief-state $b_t$ observes a frame $x_t$, produces an action $a_t$ and receives a reward $r_t$. The goal of the optimization is to maximize the discounted sum of rewards $R(b_t) = \mathbb{E}\left[\sum_{s=t}^{\infty} \gamma^{s-t} r_s | b_t\right]$ with a discount $\gamma$ and subject to entropy regularization of the policy.

The agents are trained in a distributed setting using the IMPALA framework [19], which we describe here briefly. There are N parallel 'actors' acting in an environment and collecting their experience in a replay buffer. There is one learner which takes subsets of trajectories, forms a batch and performs learning. The learning step consists of unrolling the recurrent core of the agent and computing the losses and gradients. A given piece of experience is used only once and is only slightly off policy. The policy loss and model loss are computed in the same networks, both passing gradients into the main recurrent core.

## Appendix E    Data pre-processing for plotting

For all plots showing RL scores, position accuracy and map MSE we first smooth each individual curves using an exponential moving window of length 10. We then sub-sample the curves using linear interpolation, re-evaluating them at 128 points uniformly covering the x-axis range. The shown error bars correspond to the $90\%$ confidence region.

## Appendix F    Hyperparameters

The hyper-parameter values used in experiments are reported in Table 2. In addition, Table 3 and Table 4 report the parameters for slot-based and Kanerva memory respectively. Please refer to [18] and [65] for explanations of the memory models' parameters.

| Hyper-parameter | Description | Range |
|---|---|---|
| $\mu$ | learning rate | [0.0001-0.0002] |
| $c$ | policy entropy regularization | [0.03-0.0005] |
| $\beta_1$ | Adam $\beta_1$ | [0, 0.95] |
| $\beta_2$ | Adam $\beta_2$ | [0.99, 0.999] |
| $L_o$ | Overshoot Length | {1, 2, 3, 6, 12} |
| $L_u$ | Unroll Length | [24-100] |
| $N_t$ | Number of points used to evaluate the generative loss per trajectory | [6] |
| $N_g$ | Number of points used to evaluate the generative loss per overshoot | [2] |
| $N_s$ | Number of ConvDRAW Steps | [8] |
| $N_h$ | Number of units in LSTM | [512-1024] |

Table 2: Hyper-parameters used. Each reported experiment was repeated at least 3 times with different random seeds. The reported curves for each model are the best we found with the hyper-parameters in the ranges shown above.

| Hyper-parameter | Description | Range |
|---|---|---|
| $K$ | memory size | 1350 |
| $D$ | word size | 200 |
| $n_r$ | number of reads | 3 |
| $n_k$ | top k entries for read | 50 |

Table 3: Hyper-parameters used for RMA memory.

## Appendix G    GECO

It has been shown [28] that latent variable models defining a conditional density $p_\theta(x, z) = \mathcal{N}(x|g_\theta(z), \sigma)\pi_\theta(z)$ such as ConvDRAW and VAEs can achieve better sample-quality if we constrain

| Hyper-parameter | Description | Range |
|:---:|:---|:---|
| $K$ | memory size | 32 |
| $D$ | word size | 512 |
| $\sigma_n$ | initial noise variance | 1.0 |
| write projection | linear | |
| read projection | 2-layer MLP with hidden layer size 400 | |

Table 4: Hyper-parameters used for Kanerva memory.

the reconstruction error to be not larger than a given threshold $\kappa$. A Lagrangian formulation of this constrained optimization problem can be written as a min-max optimization instead of direct ELBO maximization. More specifically, we train ConvDRAW following

$$\theta^{\star}, \phi^{\star} = \min_{\theta, \phi} \max_{\lambda \in [0, 1000]} \left[ \mathbf{KL}[q_\phi(z|x); \pi_\theta(z)] + \lambda \mathbf{E} \left[ \|x - g_\theta(z)\|^2 - \kappa \right] \right],$$

where $q_\phi(z|x)$ is an approximation to the posterior distribution $p_\theta(z|x)$ with parameters $\phi$.

We look at the effect of the choice of $\kappa$ in our belief-state model and observe empirically that there is an optimal range of $\kappa$ values with respect to map reconstruction. This analysis is shown in Figure 9.

Figure 9: Effect of the choice of GECO's $\kappa$ threshold on map-reconstruction using SimCore. We find that a value $\kappa \approx$ 1e-3 produces the best results for map reconstruction.

## Appendix H   Map decoding on Random City environment

Here we show additional map decoding samples for each of model in Figure 10.

## Appendix I   Procedurally generated terrain

We created a $64 \times 64$ procedurally generated terrain to analyze the belief state in a more naturalistic environment, Figure 11. See the caption for a description and analysis.

We also show in Figure 12 the map decoding mse and a few map decoding samples along a single trajectory.

## Appendix J   Extra levels and samples from the model

We also trained the agent on harder versions of the levels. The first level is a higher version of the cliff where the agent learns to build longer staircase. The remaining two consists of food placed in an even higher level locations, showing agent building high structures.

Figure 10: Additional map decoding samples for each model. All models were trained for the same amount of iterations. For each model we show 8 samples for overshoot length $L_o = 1$ (left) and 8 samples for $L_o = 12$ (right). We used unroll length $L_u = 100$ and the maps were extracted at time-step 70. These results confirm that the best models allow to decode the maps with substantially more details compared to other baselines.

We show a number of rollouts from the model in different environments Figure 13 and Figure 14. To make a rollout, we simulate deterministically in latent space. To obtain a frame, we sample from the convDRAW model. If the simulation knows well what should be in a given frame, the sample should match the actual frame. If it does not, either because of limitation of the model or because it has not seen that part of the environment, it should sample something consistent with its knowledge, but (most likely) different from the actual frame.

Figure 11: Terrain. The agent moves around a procedurally generated terrain. The first row in each of the three sections show the frames seen by the agent (only a fraction of frames are shown to display a large part of an episode). The second row shows top down view. Conditioned on state, we trained the decoder to predict the top down view (without passing gradients into belief state formation). The third row of each section shows the map reconstructed from the belief state. We also trained ConvDRAW as a model of the map conditioned on the belief state. In the last three rows we show samples from the model. If the agent is uncertain about parts of the map, the model should sample random pieces of map in those locations. This environment is harder than the city environment in that it is larger ($64 \times 64$), the speed of the agent is slower and it is run in RL setting (with the goal of collecting 20 randomly placed yellow blocks). We see that the agent does form a map that persist for some time but the map also fades away slowly the longer the agent does not see that part of the environment.

Figure 12: Illustration of the Map decoding MSE along a single trajectory in the procedural terrain environment. At each inset we can see from top to bottom: the true top-view, the decoded top-view and the first-person-view at the same time. This graph show that the Map MSE decreases along a single trahjectory, indicating that the model was successful at accumulating and remembering evidence about the environment's layout.

Figure 13: Inputs and samples from the building levels. First two rows show input and samples from the model. These continue into the row three and four. Then, the process repeats for another example. Top rows shows agent simulating building stairs. The second set shows the level with food placed at high location and the last set shows the agent building a tower to climb to a platform.

Figure 14: Inputs and samples from the terrain environment.

## Appendix K  SimCore python pseudo-code

```python
"""Unrolls the model starting from a random subset of agent states and
    calculates the model likelihoods at random points.
"""

import sonnet as snt
import tensorflow as tf

def swap_time_batch(t):
  return nest.map_structure(
      lambda x: tf.transpose(x, [1, 0] + range(2, x.shape.ndims)), t)

def extract_patches_1d(x, window_size):
  """Extracts 1D patches.

  Extracts 1D patches of fixed window on the time dimension.

  Args:
    x: Tensor of shape [T, ...].
    window_size: Int. Size of the window.

  Returns:
    Tensor of shape [window_size, T - window_size + 1, ...].
  """
  s = x.get_shape().as_list()
  if len(s) > 2:
    x = tf.reshape(x, (s[0], -1))
  x = tf.expand_dims(tf.expand_dims(x, 0), -1)  # [1, T, B, 1].

  batched_x = tf.extract_image_patches(
      x,
      ksizes=(1, window_size, 1, 1),
      strides=(1, 1, 1, 1),
      rates=(1, 1, 1, 1),
      padding='VALID')
  # [window, B', ...].
  batched_x = tf.transpose(tf.squeeze(batched_x, 0), (2, 0, 1))
  new_shape = batched_x.get_shape().as_list()
  return tf.reshape(batched_x, new_shape[:2] + s[1:])

def unroll_sim(core, indices, init_states, inputs, sim_length):
  """Unrolls a sim core."""

  # Returns possibly nested [D, T', B, ...]

  def f1(x):
    y = extract_patches_1d(x[1:], sim_length)
    yt = swap_time_batch(y)
    yt = tf.gather(yt, indices)
    return swap_time_batch(yt)

  staggered_inputs = nest.map_structure(f1, inputs)
  shape = nest.flatten(staggered_inputs)[0].shape.as_list()
  staggered_inputs = nest.map_structure(snt.MergeDims(1, 2),
                                        staggered_inputs)  # [D, T' *
    B, ...]

  init_states = nest.map_structure(lambda x: tf.gather(x, indices),
    init_states)
  initial_state = nest.map_structure(snt.MergeDims(0, 2), init_states)
```

```
61   sim_outputs, _ = tf.nn.dynamic_rnn(
62       core,
63       inputs=staggered_inputs,
64       initial_state=initial_state,
65       time_major=True)
66
67   resh = lambda x: tf.reshape(x, shape[:3] + x.shape.as_list()[2:])
68   sim_outputs = nest.map_structure(resh, sim_outputs)  # [D, T', B,
        ...]
69
70   return sim_outputs
71
72
73 class SimCoreHead(snt.AbstractModule):
74   """Model loss head.
75
76   From states of the trajectory, unrolls sim core at random subset of
       indices,
77     and applies generative model loss at a random index of the rollout
       .
78
79   Args:
80     core: simcore rnn core (instance of snt.RNNCore).
81     model: Class implementing a conditional generative model of frames
       .
82     sim_length: Number of steps the sim core is unrolled (overshoot
       length).
83     num_to_model_time: number of model rollouts to using (each rollout
        will start from a random point in the trajectory),
84     num_to_model_sim: how many indices along each rollout to compute
       the model's loss.
85
86   Returns:
87     loss
88   """
89
90   def __init__(self,
91                core,
92                model,
93                sim_length,
94                num_to_model_time,
95                num_to_model_sim,
96                name='model_sim_head'):
97     super(SimCoreHead, self).__init__(name=name)
98     self._core = core
99     self._model = model
100    self._sim_length = sim_length
101    self._num_to_model_time = num_to_model_time
102    self._num_to_model_sim = num_to_model_sim
103
104  def _build(self, frames, actions, states):
105    """Connects simulation and model.
106
107    Arguments are in the format: time x batch x ...
108
109    Args:
110      frames: Input frames (used for prediction)
111      actions: Actions that led to a given frame in one hot format
112      states: The belief states of an agent - the states of the rnn
       that
113         forms the belief states
114
115    Returns:
116      loss
117    """
```

```
118    seq_length = nest.flatten(states)[0].shape.as_list()[0]
119
120    # Extract indices from which to do sim rollouts
121    sim_time_indices = tf.random.shuffle(
122        tf.range(seq_length - self._sim_length, dtype=tf.int32))
123    sim_time_indices = sim_time_indices[:self._num_to_model_time]
124
125    # Unroll sim. Returns tensors of shape:
126    #   [sim_length, num_to_model_time, batch, spatial dims...]
127    sim_outputs = unroll_sim(
128        self._core, sim_time_indices, states, actions, self.
    _sim_length)
129
130    # Select states for conditioning and corresponding frames for
    modeling
131    dt = tf.random.shuffle(tf.range(self._sim_length, dtype=tf.int32))
132    dt = dt[:self._num_to_model_sim]
133    model_cond = nest.map_structure(
134        lambda x: snt.MergeDims(0, 3)(tf.gather(x, dt)), sim_outputs)
135    frame_indices = snt.MergeDims(
136        0, 2)(1 + tf.expand_dims(dt, 1) + tf.expand_dims(
    sim_time_indices, 0))
137    frames_dt = snt.MergeDims(0, 2)(tf.gather(frames, frame_indices))
138
139    # Model update
140    model_result = self._model(frames_dt, model_cond)
141    loss = tf.reduce_mean(model_result['loss'])
142
143    return loss
```

Listing 1: SimCore Loss Pseudo-Code.