[Reviews · NeurIPS 2019]

Reviewer 1



Post rebuttal update: I appreciate the additional explanation for need of overshooting in empirical methods, and the clarity of response regarding stochastic models. The issue I took was with Sec 2.2, that next-step prediction is insufficient to produce belief states, which is only an issue with approximation error when dealing with empirical results. This is not clearly explained in the paper, but clarified much more nicely in the rebuttal. This would cause me to raise my score from a 3 to a 4 for the misunderstanding, but I still do not find this paper worthy of acceptance. I don't think they are particularly surprising insights, and it seems the sole merit of this paper is an empirical one, and impressive because of performance on complex tasks.  In that case, I strongly believe that those environments and their method should be open-sourced. Getting image-based, model-based RL working is not a trivial task -- there are many tricks in the Planet paper to get their final performance, and their results are completely non-reproducible without them, and their code is open sourced. This paper is not useful to practitioners and other researchers without seeing those insights and being able to build on top of their results. -------------------------------------- While the authors present a detailed list of related work in model-based reinforcement learning, it is unclear what is novel in this work, and what the message is. They experiment with various tricks introduced in prior works like self-supervised losses, overshooting, and memory architectures and report on performance results in a handful of environments. Originality: There are new environments presented, but using existing techniques to perform a survey over combinations of techniques already shown in other work. Quality: The submission does not provide theoretical justification for their claims, but have significant results showing spread in performance across a variety of loss functions, overshoot lengths, and architectures. Clarity: The introduction is clear in the hypothesis being presented, but the experimental results Significance: This paper has low significance, as it is testing a hypothesis that stochastic generative models benefit more from overshooting than deterministic ones, while failing to theoretically disprove that next-step predictive models are insufficient for learning belief states. There are only empirical results presented showing that overshooting increases performance, but this does not necessarily mean that next step is insufficient for forming belief states, as increasing the overshoot length also increases the supervision and number of labels being used.

Reviewer 2



This paper provides an interesting comparison between different methodological approaches and nicely addresses some interesting theoretical issues associated with model conditioning for POMDP. Many of the technical details are original and the presented work seems theoretically sound. However, the clarity of the paper suffers from the amount of material covered and the whole thing could be better organised. The description of the contribution seemed to be relatively fluid throughout the paper. However after some re-organisation of the document I think this will makes a solid contribution to the community. Below I highlight specific issues and points of clarification. In the abstract, the last line, the authors say “In practice, using a expressive generative model in RL is computationally expensive and propose a scheme to reduce this computational burden allowing us to build agents that competitive model three baselines”. It is not clear where this demonstrated, please clarify. The authors confront the fact they do no consider planning algorithms and a summary of the work is “how good is the belief state at enabling standard RL algorithms”. This seems fine, as the focus is on the beliefs formed by the model, not action per se. However, the comparison to [20] is important because the focus on decoding from the belief state is covered in [20].Could the authors say a little more about the works relationship to [20] A belief-state is defined as the sufficient statistics of future states. This seems ambiguous and potentially misleading. Is it necessarily over future states, is it the sufficient statistics of the state? Is it not a probability distribution over world states? They cite [9,10] saying that a belief state is a vector representation that is sufficient to predict future observations. Again, the original statement potentially misleading. The difference between belief-state and state, i.e. SimCore starts with a belief state and then predicts a state at some future time. In the table, this is cleared up somewhat, where they say that state initialization is done such that s = b. But why the difference? Some intuition as to why ConvDRAW + GECO solves conditioning would be nice. I guess the fact that a map can reconstructed form the LSTM state is not hugely surprising. The authors note that contrastive loss is poor at mapping but good a localisation. I think theme emerging here is between local and global predictions. Could the authors comment on this. Agent trained using IMPALA, a policy gradient method. This is, presumably, for the agent core, not the simulation core. Authors state running speed decreased 20-40% compared to an agent without a model - why would this be the case? Surely the model is more of a computational burden. It seems pretty obvious why map construction should fail when there is a task because the environment is only partially sampled. I think this good example why building a map per se is not really helpful for RL in general. I really think the paper would benefit by discussing these ideas in the context of mode-based reinforcement learning. The voxel environment tin Section 4.3 is interesting but there is little information about this and it is not clear what to conclude. There is also no consideration of where this work sits in the literature.

Reviewer 3



This submission includes a number of original and significant contributions and show-cases them in different, challenging first-person environments and various experiments. Rather than training training forward models in isolation and then using them for control, the authors train them jointly with a model-free policy, using the internal state of its recurrent network as a belief state from which future frames are predicted. Hence, the paper addresses several current research topics of high interest: (1) training powerful environment models, (2) effective memory methods for control in POMDPs and (3) improving the sample complexity of model-free RL algorithms. The authors make a case for the necessity of overshooting during RL. I find the motivation insightful since overshooting is usually motivated by the desire to produce coherent multi-step predictions (for planning) rather than by improving conditioning. The experimental evaluation is clear and insightful but, as mentioned in the conclusion, leaves the reader with a few unanswered questions. In particular, it would have been nice to further investigate the interplay between policy and forward model performance. That being said, the experiments are certainly impressive, especially the ones in the voxel environment that demonstrate data efficiency, and support the main theses of the paper quite well.

[Author Response · NeurIPS 2019]

# Author Response for 'Shaping Belief States with Generative Environment Models for RL'

We are grateful to all constructive and actionable feedback provided by the reviewers. We will account for all comments and suggestions in the revised version. We are especially thankful for the detailed feedback provided by **R2** and **R3**. We believe to have addressed the key concerns raised by the reviewers below.

**General Comments: R1** expressed concerns regarding the novelty of this work, observing that most of the experiments in the paper involve published components. We also understand **R1**'s concerns with our main hypothesis as it has not been explored in the literature yet. We are working to improve our explanations in section 2.2 based on all feedback that we received.

We emphasize that careful empirical experimentation in ML can also bring valuable insights to the community. While the many ingredients of our paper exist in isolation, very little is known about how they interact with each other. As we explore complex RL agents, these different components must work together. Thus an empirical study of these interactions was timely.

As **R1** rightly observed, one of the most important contributions was to formulate and empirically evaluate a hypothesis regarding the formation of useful belief states in environment models. This involves the interaction between overshoot, probabilistic models and memory. Studying these factors require an intersectional empirical study such as this paper. Our hypothesis can be split in two parts: **H1** Overshoot is useful to learn models with long-term dependencies and **H2** Probabilistic models benefit more from overshoot than Deterministic models.

By the chain rule of probabilities we can always write the joint pdf of all observations as a product of next-step conditional pdfs, so we know that is not a problem in principle. But this is an incomplete view of the problem as it disregards the properties of the data. It is known theoretically and empirically that the properties of the data are as important as choosing the right model (e.g. imbalanced datasets require calibration, non-iid data require causal corrections). A more relevant question is: What is necessary to learn such models efficiently? We could formulate these as more formal statements about the statistical properties of the data collected by an agent walking in a 3D environment but this is beyond the scope of this paper. We expand these hypothesis here for clarity. **H1**: This is a problem of data imbalance and causal learning. Two successive frames observed by an agent walking in a 3D environment are highly correlated with each other, this implies that a model can predict the next frame with high accuracy without knowing much about the environment (e.g. by learning to displace the previous frame). However, two distant frames have a much less relation to each other (imagine two frames captured on opposing sides of a wall). In this case the probabilistic model cannot predict one frame by merely displacing the other, it is necessary to use a more global representation of the environment for such predictions. **H2**: With overshooting, another problem emerges: Due to the partial observability of the environment, the entropy of future frames conditioned on the past grows with the overshoot length. It is clear that deterministic models cannot perform multi-modal predictions. For sufficiently long overshoots, any deterministic prediction will inevitably converge to the average of all possible frames that could be seen. Since this prediction is independent of the belief-state, deterministic models should not benefit from long overshoots. Our experiments provide strong evidence for **H1** and **H2** on numerous complex environments.

**Overshoot just adds more labels:** Our experiments reject this hypothesis. If this was the case, increasing the overshoot length would not affect the asymptotic error in top-down view reconstruction, only the convergence time. As we can see in Figures 3 and 4 this is not the case.

**Relevance to RL:** We wholeheartedly agree that doing planning with our models would be a great follow up work. But we also believe that our experiments show benefits to the performance of a strong baseline model-free RL agent in complex environments due to the model. We appreciate the suggestion and will expand the discussion about model-based RL.

**Why a new environment?** We appreciate the observation and will expand a lot more the details in the revision. One of the main reasons for a new environment was that we believe that the benefits of expressive belief-state models will become more evident in combinatorial and compositional environments, where the agents are expected to perform a variety of different tasks. Our experiments indicate a substantial data-efficiency gain in these environments, Figure 7, and also add more evidence to the hypothesis put forward in the paper.

**Comparison to [20]:** Indeed, reference [20] is the closest to our paper. For this reason, we dedicated an entire paragraph in section 3 discussing our innovations relative to [20]. Some of the key components of our paper, namely the interaction between overshoot and generative models are not addressed in [20].

**It is not surprising to decode the top-view from the LSTM state:** Learning to extract the top-view of an environment uniquely from actions and first-person-views is by far not a trivial problem. As discussed in section 2.3, all known solutions involve a substantial injection of prior knowledge in the models.

[Meta-Review · NeurIPS 2019]

This paper examines the use of generative models for developing representations to improve data efficiency in RL. Specifically, the authors use a generative model that is trained to predict multiple frames into the future (overshooting), and they show that when the model is stochastic (but not deterministic), overshooting leads to useful representations of the environment that can improve RL efficiency. The reviews on this paper were fairly divergent in the first round. Two of the reviewers liked this paper, but one did not feel it provided truly novel contributions, and only brought together previously proposed ideas for using predictive training to improve RL representations. In discussion, the reviewers came to the conclusion that it does demonstrate the utility of overshoot prediction for stochastic models and that an empirical demonstration like this can be useful. But, it was also agreed in discussion that open code would provide means for others to verify and build on these empirical results. In the end, it was decided by the AC and SAC that this paper makes sufficient contribution to be accepted at NeurIPS, but given the discussions, we would strongly encourage the authors to release any code they can. For empirical demonstrations of the utility of ideas like this, open code can greatly enhance the impact of the contributions.